# Flow Cytometric Identification of Hematopoietic and Leukemic Blast Cells for Tailored Clinical Follow-Up of Acute Myeloid Leukemia

**DOI:** 10.3390/ijms231810529

**Published:** 2022-09-11

**Authors:** Vera Weeda, Stefan G. C. Mestrum, Math P. G. Leers

**Affiliations:** 1Department of Clinical Chemistry & Hematology, Zuyderland Medical Centre, 6162BG Sittard-Geleen, The Netherlands; 2Department of Molecular Cell Biology, GROW-School for Oncology and Reproduction, Maastricht University Medical Centre, 6200MD Maastricht, The Netherlands

**Keywords:** hematopoietic stem cells, leukemic stem cells, flow cytometry, distinction, CD marker expression profiles

## Abstract

Acute myeloid leukemia (AML) is a myeloid malignancy that is characterized by the accumulation of leukemic blast cells, which originate from hematopoietic stem cells that have undergone leukemic transformation and/or are more mature progenitors that have gained stemness features. Currently, no consensus exists for the flow cytometric identification of normal blast cells and their leukemic counterparts by their antigenic expression profile. Differentiating between the benign cells and the malignant cells is crucial for the further deployment of immunophenotype panels for the clinical follow-up of AML patients. This review provides an overview of immunophenotypic markers that allow the identification of leukemic blast cells in the bone marrow with multiparameter flow cytometry. This technique allows the identification of hematopoietic blast cells at the level of maturing cells by their antigen expression profile. While aberrant antigen expression of a single immunophenotypic marker cell cannot be utilized in order to differentiate leukemic blast cells from normal blast cells, combinations of multiple immunophenotypic markers can enable the distinction of normal and leukemic blast cells. The identification of these markers has provided new perspectives for tailored clinical follow-up, including therapy management, diagnostics, and prognostic purposes. The immunophenotypic marker panels, however, should be developed by carefully considering the variable antigen marker expression profile of individual patients.

## 1. Introduction

Acute myeloid leukemia (AML) is a type of hematological malignancy that typically arises in the bone marrow (BM), and it currently accounts for approximately one-third of all myeloid malignancies in adults. Therefore, AML is the most frequently encountered type of acute leukemia in adults [1]. The incidence of AML increases with age, meaning that the increasing life expectancy leads to a higher overall incidence of AML [1,2].

AML is defined in the 2017 WHO classification as an increase in myeloid blast cells, typically exceeding 20% of all of the BM cells [3]. When these malignancies are induced by their mutational background, the normal blast cells undergo leukemic transformation and/or become more mature progenitors that have gained stemness features [2,3]. These cells are the so-called leukemic blast cells. The accumulation of these leukemic blast cells causes the expulsion of healthy hematopoietic cells in the BM. Without intervention, AML leads to possibly life-threatening symptoms (e.g., anemia, infections, and bleedings), resulting in severe comorbidities and a high mortality [2].

In the last 10 years, novel treatment strategies that facilitate a personalized approach for AML have been widely investigated in (pre-)clinical studies and clinical trials (e.g., FLT3 inhibitors, a combination of hypomethylating agents and Venetoclax, CPX 351, and IDH1/IDH2 inhibitors) [4]. However, AML patients are still primarily treated with chemotherapy, which eliminates the leukemic blast cells by toxicity and the induction of apoptosis. In addition, the eligibility of AML patients for allogeneic stem cell transplantation is based on their age, their overall fitness, and the cytogenetic and mutational risk stratification, as recommended by the European LeukemiaNet [5].

The very poor overall survival rates (the 5-year survival rate is 25%) and the high relapse rates after chemotherapy are still a major challenge to overcome [2,6,7]. Approximately 50% to 75% of adults who are diagnosed with de novo AML achieve complete remission after chemotherapy. Despite these high numbers of complete remission, only 20% to 30% of patients remain disease-free in the long term [8]. The low long-term disease-free survival rate indicates that part of the leukemic blast cell population is resistant to the effects of chemotherapy and, therefore, a more tailored therapeutic approach could be beneficial.

The increasing incidence, the poor overall survival rate, and the high risk of relapse of AML suggest that future research that focusses on personalizing the clinical follow-up at the individual patient level is essential. Since AML is a malignancy of the blast cells, expanding the knowledge about characterizing the normal blast cells and their leukemic counterparts is crucial in order to develop new diagnostic strategies.

Currently, determining if chemotherapy has eliminated all of the leukemic blast cells is challenging. The appearance of new normal blast cells during and after chemotherapy treatment is often confused with the presence of leukemic blast cells, since these cells display (almost) identical morphological characteristics. Furthermore, the distinguishment of the normal blast cells and the leukemic blast cells based on their immunophenotype is challenging with the existing flow cytometric immunophenotype panels. Moreover, the present minimal residual disease (MRD) quantification (which is based on multiparameter flow cytometry and molecular analyses) may be improved by the identification of highly specific leukemic blast cell markers [9]. The improper identification of leukemic blast cells can result in inadequate treatment, allowing the remaining leukemic blast cells to expand, which ultimately leads to relapse. In addition, the overestimation of the number of leukemic blast cells by cytomorphology could lead to unnecessarily prolonged treatment, increasing the risk of more severe treatment-related complications. Therefore, more accurate differentiation between the healthy normal blast cells and the leukemic blast cells is crucial for a more personalized clinical follow-up of AML.

With the efforts of the EuroFlow consortium and the European LeukemiaNet, the use of antibodies for the immunophenotyping of leukemia and lymphoma was standardized, which resulted in antibody panels that allow the phenotyping of the dominant blast cell population in the BM [10,11]. These panels, however, can only identify the general leukemic blast cell population, without differentiating between the different blast cell subsets [10,12].

The knowledge about the antigen expression profiles of the normal and the leukemic blast cells in AML patients has been expanded immensely by recently conducted multiparameter flow cytometric studies of these cells [13,14,15]. Although these profiles were studied extensively, no consensus exists for the identification of normal blast cells and their leukemic counterparts by their antigen expression profile with multiparameter flow cytometry. Furthermore, some of the described leukemic markers in the literature are also expressed on healthy hematopoietic cells, which underlines that more sophisticated flow cytometric antibody panels are required for the proper identification of leukemic blast cells.

The definition of the healthy blast cells or the leukemic blast cells with flow cytometry is, usually, exclusively possible because of an expansion of the blast compartment (at the onset) or if the aberrant markers are co-expressed by the blasts. Even identifying the maturation step that is represented could be unhelpful in assessing whether blast cells are healthy or malignant. It is, therefore, of paramount importance to identify the markers that are not expressed in the normal blast cell population at any level of their hierarchy, while understanding from which level of the hierarchy these branching leukemic blast cells develop and which markers are exclusively expressed on these cells is ancillary for the patient management. Structuring this knowledge can improve the understanding of how the normal blast cells and their leukemic counterparts can be distinguished by their antigen expression profiles, therefore enabling the development of novel flow cytometry panels.

These panels can potentially advance the diagnostic work-up of AML by generating new approaches for, among other factors, the follow-up after treatment. In this review, we aim to give an overview on how to distinguish the normal blast cells from their leukemic counterparts in the BM of these patients based on their immunophenotype.

## 2. Human Blast Cell Hierarchy

The hematopoietic stem cells (HSCs) are a self-renewing blast cell population that can eventually develop into all types of mature blood cells, and are able to repopulate the BM if it is necessary [16]. These cells reside at the apex of the hierarchy of the hematopoietic progenitor cells.

The maturation and differentiation of HSCs into mature hematopoietic cells can be presented in a hierarchical structure, as shown in Figure 1. The CD34-negative HSCs reside at the apex of this hierarchy. These cells are capable of self-renewal and are thought to behave as long-term repopulating cells. These cells remain quiescent until the BM is significantly depleted of its cells, which consequently causes these cells to proliferate in order to repopulate the BM. These CD34-negative HSCs then differentiate to CD34-positive HSCs, which are also capable of self-renewal. The CD34-positive HSC population comprises long-term repopulating cells, as well as short-term repopulating cells [16,17]. The CD34-negative HSCs also show megakaryocyte and erythrocyte differentiation potential and can, therefore, differentiate to megakaryocyte-erythrocyte progenitors (MEPs) via a bypass route [17].

The processes of quiescence, proliferation, and differentiation are tightly regulated in the short-term CD34-positive repopulating cells in order to supply the BM with sufficient progenitor cells and ultimately mature cells. These cells mature into multipotent progenitors (MPPs), which are still not limited in their differentiation capacity by lymphoid-commitment or myeloid-commitment [18,19]. The MPPs, in turn, differentiate into two types of progenitors, which are the common myeloid progenitor (CMP) and the lymphoid-primed multipotent progenitor (LMPP) [20,21,22]. The outcome of this differentiation step determines, for the majority of cells, whether they will ultimately become mature myeloid cells or lymphoid cells.

The CMPs can directly mature into basophils or eosinophils [23], but can also develop into MEPs and the granulocyte/monocyte/dendritic cell progenitor (GMDP) [18,24,25]. Subsequently, the MEPs differentiate to erythrocytes or megakaryocytes [26], while the GMDPs mature into monocyte/dendritic cell progenitors (MDPs), basophils, eosinophils, or neutrophils [27,28]. The MDPs can then develop into the common dendritic cell progenitor (CDPs) or into monocytes and, ultimately, to macrophages [17,28]. In the case that cells mature into CDPs, these cells will further mature into dendritic cells [25,28,29].

The MPPs can also differentiate to the lymphocytic precursor LMPP, which gives rise to the multi-lymphoid progenitor (MLP) [25]. Similarly to the GMDP, an MLP can also differentiate to an MDP [25], which in turn matures into the early T-cell progenitor (ETP), followed by further maturation into the mature T-cell [20,21]. The MLPs can also differentiate to a pro-B cell, which differentiates to the pre-B cell and, ultimately, to the immature B cell [30].

### 2.1. Hematopoietic Stem Cell

Recent literature describes two distinct subsets of HSCs, based on their CD34 expression status (CD34-negative and CD34-positive HSCs) [16,17]. The differentiation of these cell populations is possible throughout their differential antigen expression profile. Table 1 summarizes the expression profiles of the antigens for the HSCs and the subsequent progenitor cells.

The CD34-negative HSCs show low levels to a completely absent expression of CD38, although these cells are positive for CD90 [18,19]. The lack of CD34 expression distinguishes the CD34-negative HSCs from the other (CD34-positive) HSCs and progenitors. The CD34-negative HSCs are also negative for CD45RA, CD110, and CD135 (also known as FLT3) [17,31,32].

The more mature lineage-committed blast cells are also known for their CD34-negative phenotype, which creates challenges for differentiating the lineage-committed blast cells from the more primitive CD34-negative HSCs. However, the CD34-negative HSCs are CD133-positive and GPI-80-positive, while the lineage-committed blast cells show no expression of these markers [17,31,32].

When the CD34-negative HSCs undergo the process of maturation, these cells mature into CD34-positive HSCs [19]. This cell compartment is still negative for CD38 and remains positive for CD90, CD133, and GPI-80 [19,33,34]. In addition, the CD34-positive HSCs also express CD44 and CD135, which are not present on their more primitive counterparts [34,35]. The antigens that are confirmed to not be expressed by the CD34-positive HSCs are as follows: CD3, CD10, CD11c, CD14, CD19, CD45RA, CD56, CD66b, and CD335 [17,18,25,33,34].

### 2.2. Multipotent Progenitor

As the CD34-positive HSCs mature, these cells eventually differentiate to MPPs. Similarly to the CD34-positive HSC, the MPP is located in the CD34-positive/CD38-negative compartment [19]. During the differentiation of the CD34-positive HSCs to MPPs, the cells maintain the expression of CD44, CD133, and CD135 (FLT3), while expression of CD90 is ceased [33,34,36,37]. Furthermore, these cells do not express CD45RA and CD49f. The other markers that are not expressed by MPPs are as follows: CD3, CD10, CD11c, CD14, CD19, CD56, CD66b, CD90, CD335, and GPI-80 [17,33,34,38].

### 2.3. Common Myeloid Progenitor

When the MPPs differentiate to the more mature CMPs these cells acquire the expression of the CD38 antigen [19]. After this maturation step, all of the subsequent myeloid progenitor cells belong to the CD34-positive/CD38-positive blast cell compartment. The CD38 is a multifunctional protein that functions as an enzyme and studies on animals indicate that CD38 expression protects against infection [39]. Based on their CD34-positive/CD38-positive phenotype, these cells can be distinguished from the more primitive MPPs and HSCs, which do not express CD38. The CMPs show intermediate expression levels of CD123, while the expression of this protein is absent in the more primitive blast cell compartments [34,40]. Similarly to the CD34-positive HSCs and MPPs, the CMPs also express CD135. The CMP is otherwise characterized by the negativity of several markers, including the following: CD3, CD10, CD11c, CD14, CD19, CD45RA, CD56, CD66b, and CD90 [17,18,25,34,41].

### 2.4. Megakaryocyte-Erythrocyte Progenitor

The CMP then matures into more committed progenitor cells. One of these progenitor cell types is the MEP [19]. Intermediate levels of CD123 expression are found on MEPs, which are similar to the expression levels that are found on the CMPs [40]. As concluded by Radtke et al., CD133 is also expressed in low levels on the MEP. This is the only blast cell subpopulation besides the CD34− HSCs that does not express CD135, allowing the distinction of MEPs from CD34+ HSCs and the other myeloid progenitor cell populations. The MEPs also lack the expression of the following other markers: CD3, CD10, CD11c, CD14, CD19, CD45RA, CD56, CD66c, and CD90 [17,18,25,34,41,42,43].

### 2.5. Granulocyte/Monocyte/Dendritic Cell Progenitor

Compared to the previous myeloid progenitor cell types, the GMDPs express many other markers, which include CD33, CD45RA, CD117, and HLA-DR. The expression of CD135 on the GMDPs is similar to that of the HSCs, the MPPs, and the CMPs [28,29,34,40]. As opposed to the CMPs and the MEPs, the GMDPs are negative for CD123 [28]. The other markers that the GMDPs lack the expression of include the following: CD1c, CD3, CD11c, CD14, CD19, CD56, CD66b, CD115, and CD116 [18,25,28,29,34].

### 2.6. Monocyte/Dendritic Cell Progenitor

Similar to the GMDP, the MDP expresses CD33, CD45RA, CD135 (FLT3), CD117, and HLA-DR. The MDP also expresses CD115, which is not expressed on all of the previously mentioned progenitor cell populations [19,28,29,40]. The negativity of various other CD antigens can be utilized for the identification of these MDPs. These CD antigens comprise the following: CD3, CD10, CD14, CD19, CD56, CD66b, CD116, and CD123 [17,18,19,25,28,34]. The MDP also has been shown to exhibit an absence to low levels of CD11c [28,29].

### 2.7. Common Dendritic Cell Progenitor

The MDPs may differentiate to CDPs, which ultimately mature into dendritic cells. The CDP expresses markers that are also encountered on the MDP, such as CD33, CD45RA, CD117, CD135 (FLT3), and HLA-DR [28]. Next to those markers, the CDPs can be identified based on their CD11c-positive, CD116-positive, and CD123high phenotype. Both the CMP and the MEP express CD123, but not on a high level. The CDPs exclusively express CD11c, which allows the identification of the CDP by CD11c positivity and the CD123high phenotype [18,28,29,40]. The antigens that are not expressed on CDPs are as follows: CD3, CD10, CD14, CD19, CD56, CD66b, and CD115 [17,18,25,28,34].

## 3. Leukemic Blast Cell Markers

Generally, the leukemic blast cells that are in AML were originally thought to be exclusively located in the CD34-positive/CD38-negative (CD34+/CD38-) compartment. However, more recent evidence suggests that only the leukemia-initiating cells are located in the CD34+/CD38− compartment, while the (more proliferative) bulk of the blast cells is located in the CD34+/CD38+ compartment [44]. Although not all leukemic blast cells are CD34+/CD38-, this compartment is considered to be the most clinically relevant blast cell compartment. The CD34+/CD38− blast cell compartment contains the most chemotherapy-resistant and the least immunogenic leukemic blast cells in vitro and in vivo. Considering their CD34 and CD38 expression profile, these cells might resemble the more primitive blast cells that are known for their quiescent nature [45]. Because the quiescent blast cells exhibit low levels of proliferative activity and high levels of anti-apoptotic potential, these cells are highly resistant against chemotherapy. Furthermore, the high anti-apoptotic potential of the CD34+/CD38− leukemic blast cells prevents these cells from undergoing apoptosis as a consequence of their immunogenicity [46].

Numerous studies have been conducted in order to identify candidate antigens that allow the distinction of normal and leukemic blast cells. In Table 2, the markers that have been identified to be aberrantly expressed on the leukemic blast cells in AML and the frequency of these aberrancies are shown.

The markers CD33, CD44, CD45RA, CD99, CD123, and CD135 (FLT3) are expressed in the majority of AML cases [37,47,48,49,50,51,52,53,54,55,56,57]. However, several studies indicate that these markers are also expressed in the normal blast cells [10,40,44,50,51,52]. Therefore, these markers are not useful to distinguish the normal blast cells from their leukemic counterparts.

Flow cytometric studies have shown that CD2 is highly expressed on the leukemic blast cells in AML. However, this high expression of CD2 is not highly frequent in AML patients. In the study of Zeijlemaker et al., the expression of CD2 was detected in 18% of AML patients, in which the BM samples of 236 AML patients were subjected to flow cytometric immunophenotyping [37]. The low frequency of CD2 expression on leukemic blast cells was confirmed in a subsequent study that was conducted by Chen et al. [58]. However, in the patients with AML with mutated *NPM1*, aberrant CD2 was more frequently encountered [59]. CD2 expression is also found in normal BM on T cells and NK cells [44].

**Table 2 ijms-23-10529-t002:** Leukemic blast cell markers, the percentages of patients that the markers are expressed in, and expression of these markers in normal blast cells.

Leukemic Blast Cell Antigenic Marker	Expression in Leukemic Blast Cells (%: AML)	Expression in Normal Blast Cells	Reference(s)
CD2	18	No	[37,44,58,59]
CD7	43	No	[37,60]
CD11b	55	No	[37,40]
CD19	8	No	[44,61]
CD22	51	No	[37,44]
CD25	25	No	[44,62,63]
CD33	82–89	Yes	[44,47,48]
CD44	90–100	Yes	[50]
CD45RA	85–90	Yes	[44,51,52]
CD47	100	No	[64,65,66]
CD56	32–48	No	[37,67]
CD96	33–44	No	[44,63,68]
CD99	82–90	Yes	[44,53]
CD123	63–82	Yes	[44,54,55]
CD135 (FLT3)	54–92	Yes	[44,56,57]
CD157	97	No	[69,70]
CD244	95–98	No	[48,71,72]
CD366 (TIM-3)	62–79	No	[37,48,73,74,75]
CLL-1	85–92	No	[76,77,78]
IL1RAP	80	No	[44,79,80]
MPO	61	No	[81,82,83]
NANOG	28	No	[84]
OCT4	36	No	[84]
SOX2	35	No	[84]
SSEA1	40	No	[84]
SSEA3	28	No	[84]

Abbreviations: AML: acute myeloid leukemia.

Although CD7 is expressed on T cells and NK cells, it is not expressed on the normal blast cells or the progenitor cells in healthy BM [60]. A study investigating the CD7 expression levels on CD34+/CD38− leukemic blast cells in the BM samples of 236 AML patients concluded that CD7 was aberrantly expressed in 43% of the investigated cases. This underlines the potential of this marker for the differentiation between the normal and the leukemic blast cells [37].

CD11b is generally expressed on many leukocytes, including eosinophils, mature monocytes, neutrophils, granulocytes, macrophages, and NK cells [40]. It was also found to be aberrantly expressed on the CD34+/CD38− leukemic blast cells in 55% of AML patients [37].

CD19 is expressed on the B cells in healthy BM and functions as the dominant signaling component of a multimolecular complex on the surface of these cells [61]. While this antigen is not expressed on the healthy blast cells, it is aberrantly expressed on the leukemic blast cells in approximately 8% of AML cases [44].

CD22 is an antigen that has not been extensively investigated on the leukemic stem cells in AML. However, the study of Zeijlemaker and colleagues showed that CD22 was aberrantly expressed on the CD34+/CD38− leukemic blast cells in 51% of AML patients [37]. CD22 is also expressed on B cells in normal BM [44].

CD25 is an antigen that is normally expressed on activated B and T cells [44]. In AML, CD25 was found to be expressed on the leukemic blast cells. However, only a minority of the AML patients displayed the aberrant expression of CD25, as this antigen is encountered in approximately 25% of AML cases. In the AML cases that displayed an aberrant expression of CD25, high levels of this protein have been found, as shown in the immunophenotyping studies of the BM of AML patients [62,63].

CD47 is expressed on various hematopoietic cell lineages, including macrophages, NK cells, dendritic cells, T cells, and B cells [64]. Several studies have concluded that CD47 is aberrantly expressed in all of the leukemic blast cell compartments and in all AML patients [64,65,66]. The study of Ponce et al. found no differential expression of CD47 at the initial diagnosis, as compared to the moment of relapse [65].

CD56 is expressed on the NK cells, the αβ T cells, and the γδ T cells in normal BM. Although this protein is not expressed on normal blast cells or myeloid progenitor cells, several studies have observed that CD56 is expressed on the leukemic blast cells of AML patients [37]. In particular, the aberrant expression of CD56 was encountered on acute myelomonocytic leukemia and acute monoblastic/monocytic leukemia. CD56 seems to be expressed more frequently in AML with t(8;21) (65% of cases), as compared to other subtypes [67].

The aberrant expression of the CD96 antigen (also known as TACTILE-1) was encountered in the BM and the peripheral blood of 33% to 49% of AML cases, as was shown in two recent studies [37,63]. The expression levels of this antigen in individual AML patients varied from intermediate to high. In normal BM, CD96 is exclusively expressed in low levels on activated T cells and NK cells [44,68].

CD157 is an antigen that is mainly expressed on normal mature myeloid cells. Its expression increases when the blast cells mature into promyelocytes [69,70]. Krupka et al. studied the peripheral blood and the BM of 101 newly diagnosed or relapsed AML patients by flow cytometric analyses. This study showed that 97% (98/101) of the AML patients displayed an expression of CD157. The highest mean expression of this protein was found in the patients who were suffering from acute myelomonocytic leukemia and acute monoblastic/monocytic leukemia [70].

The marker CD244 was initially thought of as a key regulator of natural killer cells [71]. However, subsequent investigations have found that CD244 is also expressed on the γδ T cells, the basophils, the monocytes, the dendritic cells, and subtypes of the CD8-positive T cell [71]. Recent studies have concluded that CD244 is highly expressed on leukemic blast cells in most AML patients (95–98%) at the initial diagnosis, as well as in relapsed patients [48,72].

CD366, or TIM-3, is a receptor that is normally expressed on the interferon gamma-producing T cells, the FoxP3-positive regulatory T cells, and the innate immune cells (such as the macrophages and the dendritic cells) [73]. While TIM-3 is not expressed on the normal blast cells, most leukemic blast cells express this protein [74]. At the initial diagnosis, TIM-3 was expressed in 62% to 79% of the AML patients [37,48]. Furthermore, TIM-3 was expressed in around 65% of the AML patients after relapse [48]. The TIM-3 expression was shown to be particularly upregulated in the AML NOS without maturation and acute myelomonocytic leukemia, while no overexpression was observed in APL with PML-RARA [75]. TIM-3 is known to induce a wide variety of tumor-promoting mechanisms. These include the promotion of tumor progression (e.g., the facilitation of tumor cell migration and invasion), the suppression of CD4-positive T cells, and the activation of the mTOR function that stimulates the proliferation of AML cells [73].

One of the most researched CD antigens that is aberrantly expressed on the leukemic blast cells in AML is CD371 (C-type lectin-like molecule-1 or CLL-1) [76,77,78]. CLL-1 is a C-type lectin-like receptor that contributes to the regulation of the immune response by the recognition of pathogen-associated and damage-associated molecular patterns [77]. Normally, CLL-1 can be found on the mature myeloid cells and the lymphoid cells, while this marker is not expressed on the blast cells [78]. CLL-1 was found to be aberrantly expressed on 85–92% of AML cases [76].

IL1RAP is an antigen that is expressed on the T cells, but not on the normal blast cells [44]. However, several studies that have investigated the expression of this marker on the leukemic blast cells found that IL1RAP was upregulated on these cells. This upregulation is observed on the leukemic blast cells in the BM, as well as in the peripheral of the majority of the AML patients (80%) [79,80]. In the CD34+/CD38− leukemic blast cell population, an intermediate to high expression of IL1RAP was found [80].

Myeloperoxidase (MPO) is mostly expressed in the cytoplasm of the normal hematopoietic blast cells and the granulocytes [81]. In addition, MPO is expressed in varying intensities in the leukemic blast cells of AML [81,82,83]. In these studies, it has been shown that the majority of AML patients displayed an expression of MPO (61% of the studied cases).

In a recent study that was conducted by Picot et al., the expression of the embryonic antigens OCT4, NANOG, SOX2, SSEA1, and SSEA3 were investigated in the CD34+/CD38− blast cell compartment, as well as in the CD34+/CD38+ blast cell compartment, in the BM samples of 10 healthy individuals and 103 AML patients. An upregulation of the transcription factors OCT4 and SOX2 was observed in the leukemic blast cells, as compared to the normal BM cells [84]. In the CD34-positive leukemic blast cell fraction, OCT4, NANOG, SOX2, SSEA1, and SSEA3 were found to be positive in approximately 36%, 28%, 35%, 40%, and 28% of the cases, respectively [84].

## 4. Distinction between Healthy Hematopoietic and Leukemic Blast Cells

Only half of the antigens that are mentioned in Table 2 are expressed in more than 50% of AML cases. The (possibly aberrant) expression of these markers in the majority of AML patients can be useful in order to accurately identify the leukemic blast cells in these patients, and to distinguish them from the normal blast cells. The markers that are often expressed on the leukemic blast cells include the following: CD33, CD44, CD45RA, CD99, CD123, and CD135 (FLT3). However, these antigens are also found to be expressed on the healthy hematopoietic stem cells and progenitor cells. The antigens that are expressed on the leukemic blast cells in the majority of AML patients, but not on hematopoietic stem cells or the progenitor cells, include the following: CD47, CD157, CD244, CD366 (TIM-3), CD371 (CLL-1), and IL1RAP. These markers are not exclusively expressed on the leukemic blast cells; they are also encountered on several differentiated cell lineages. Therefore, differentiating the normal and the leukemic blast cells based on the analysis of a single marker is not possible. Combining these highly distinctive markers in more elaborate antigen panels, however, can be a valuable strategy for differentiating the leukemic blast cells from their non-leukemic counterparts.

A prerequisite for a proper distinction of the normal and the leukemic blast cells is that the marker profile of the normal BM cells is known, especially for the stem cell and progenitor cell compartments. Future studies are necessary in order to further define the marker profiles for these BM cell populations. The establishment of the marker profile for the healthy BM cells could also be beneficial for determining the origin of the leukemic blast cells, which allows for more elaborate subtyping of AML, and ultimately can lead to tailored flow cytometry panels for the routine diagnostic work-up of the different AML subtypes.

These tailored flow cytometry panels can be used for multiple purposes (Figure 2), including the following: for a clinical diagnosis of AML; for the distinction of the normal and the leukemic blast cells in order to estimate the MRD; for providing an ‘immunophenotypic fingerprint’ of the present leukemic blast cells, allowing immunophenotype-based risk stratification [85] and for determining the response to therapy.

### 4.1. Combination of Flow Cytometry with Other Single-Cell Techniques

The deployment of multiparameter flow cytometry panels in the routine diagnostic work-up of AML has some drawbacks. Most of the antigen markers are not exclusively expressed in 100% of the AML patients. Besides, the marker expression profiles differ between the specific AML subtypes. Therefore, the optimal antigen marker expression profiles for the different subtypes of AML, as defined by the WHO, should be studied before their implementation. By working in a stepwise approach, tailored flow cytometry panels for each of the different AML subtypes can be developed. As a first step, the acute leukemia orientation tube (ALOT) can be used in order to determine whether the patient suffers from an acute leukemia of a myeloid or a lymphoid origin [10]. After the ALOT concludes that the acute leukemia is of a myeloid origin, the BM sample of the AML patient should be subjected to immunophenotypic, cytogenetics, and molecular diagnostics in order to identify the AML subtype, according to the WHO classification. The distinction and immunophenotypic profile of the leukemic blast cells with highly specific markers (based on the AML subtype) can then be utilized for further monitoring of the individual patient.

Although flow cytometry is able to identify rare cell populations, as has been demonstrated in MRD studies by the European LeukemiaNET MRD Working Party [9] that single-cell RNA or protein analyses may be more suitable in this context. These single-cell analyses include, among others, the deployment of CyTOF, single-cell RNA sequencing, TARGET-sequencing, and proteogenomics [86,87]. For instance, van Galen et al. combined single-cell RNA-sequencing and genotyping in order to profile 38,410 cells, followed by using a machine learning classifier in order to distinguish the healthy blast cells from their leukemic counterparts [88]. In another study, CyTOF was used in order to simultaneously distinguish between the healthy and the leukemic blast cells, to analyze the functional characteristics of these cells, and to correlate these parameters with the therapeutic responses of AML patients [89]. Furthermore, the study of Dillon et al. highlighted that personalized single-cell proteogenomics can be used in order to distinguish between AML and non-malignant clonal hematopoiesis [90].

Combining flow cytometry with these techniques in order to detect (therapy specific) protein or gene markers can lead to the development of novel targeted therapy approaches, the monitoring of the initial treatment response, and the detection of MRD [91]. In our opinion, initial monitoring by flow cytometric analyses and single-cell analysis techniques for the assessment of MRD should supplement each other in order to advance the clinical management of AML.

### 4.2. Therapeutic Implications

The first targeted therapy that was based on the surface antigen expression for AML was an anti-CD33 monoclonal antibody-drug conjugate named gemtuzumab ozogamicin, which was developed in 2000, and was withdrawn from the market in 2010. However, novel insights regarding the dosing, the frequency, the response rates, and the toxicities led to the reapproval of this targeted therapy in 2017, as these insights improved its efficacy. Burnett et al. confirmed this improved efficacy by subjecting 788 AML patients to 3 mg/m^2^ or 6 mg/m^2^ within the first course of induction therapy. These treatments resulted in an overall response of 89% and 86%, respectively. Currently, novel therapies are studied, in which the chemoresistant leukemic blast cells are eliminated by targeting the antigen markers via bispecific T cell-engaging antibody constructs, in which one single-chain variable fragment (scFv) binds to the CD3 receptor of a T cell and the other scFv binds to a tumor-specific molecule on the leukemic blast cells [48]. A recently published clinical trial has also shown that the AML patients who did not respond to induction therapy responded favorably to flotetuzumab, which is a bispecific DART antibody-based molecule to CD3ε and CD123 [92]. This study concluded that 26.7% of the patients achieved complete remission, while the overall response rate (including partial hematological remission) was 30.0%. In the phase one clinical trial that was conducted by Vey et al., the AML patients were treated with a recombinant anti-CD44 immunoglobulin G1 humanized monoclonal antibody (RG7356) [93]. However, a response was observed in only 3 out of 44 patients who were treated with RG7356, which did not support the use of this antibody as a monotherapy. As this antibody showed a favorable cytotoxicity profile, combination therapy with cytotoxic agents, such as cytarabine, or in another clinical setting (e.g., consolidation or maintenance therapy) is yet to be investigated.

Considering the success of chimeric antigen receptor (CAR) T cell therapy in B-ALL, the translation of this novel therapy in AML may be a promising approach in order to increase the response and overall survival rates of these patients [94,95,96]. The identification of suitable target antigens in AML, however, is difficult, due to the low number of antigens that are exclusively expressed on the leukemic blast cells. Moreover, the expression of antigens can evolve during the progression of AML [48]. Therefore, the combinatorial targeting of antigens with a non-overlapping expression on normal cells with CAR T cells can enhance the therapeutic efficacy without excessive toxicity [48,97]. A recent study of Warda et al. concluded that CAR T cells targeting IL1RAP induced apoptosis effectively in the leukemic blast cells. They also concluded that this treatment strategy will be investigated further in a phase one clinical trial [98].

## 5. Conclusions

Promising multiparameter flow cytometry markers for the distinction of normal and leukemic blast cells were identified, which enable a more specific clinical diagnosis of AML, MRD detection, and more personalized treatment (by novel therapies). These markers have been shown to be highly specific for leukemic blast cells, and they differ from the expression profiles of the normal blast cells and the other benign cell populations in AML patients. However, these panels should be used with caution, considering that the antigen marker expression profiles vary between patients, between the different AML subtypes, and may differ before and after receiving therapy in individual patients. However, future studies should elucidate the specific marker combinations for differentiating between the normal and the leukemic blast cells in the different WHO subtypes of AML.

## Figures and Tables

**Figure 1 ijms-23-10529-f001:**
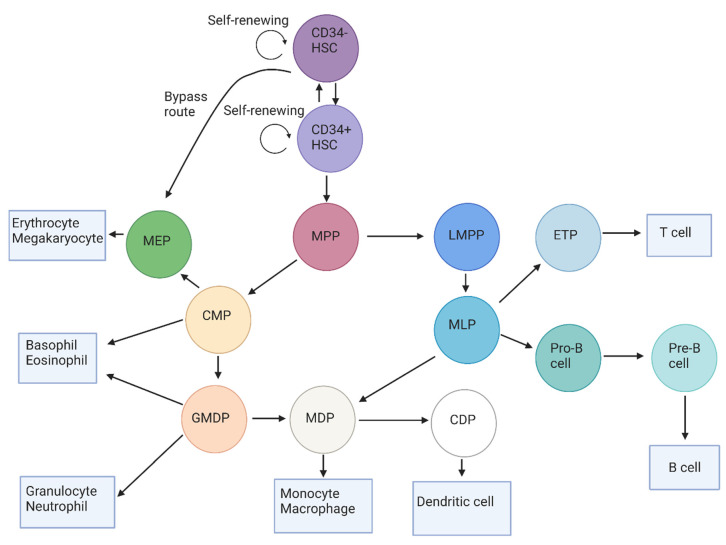
The human blast cell hierarchy, which displays the maturation of HSCs and the subsequent blast cell subtypes in the BM. At the apex of this hierarchy, the CD34-negative HSC resides, which differentiates to the CD34-positive HSC or to the MEP through a bypass route. The CD34-positive HSC further differentiates to MPPs, which are capable of maturing to either the lymphoid-committed LMPP or the myeloid-committed CMP. LMPPs then mature inro MLPs that can, in turn, differentiate to MDPs, pro-B cells, or to the ETPs. These pro-B cells ultimately mature inro B cells, while the ETPs mature into T cells that migrate to the lymphatic system. The myeloid-committed CMP can differentiate to mature basophils and eosinophils, but also to more mature blast cells (such as MEPs and GMDPs). After differentiation to MEPs, these cells either commit to the erythroid or megakaryocytic cell lineages as they further differentiate. The GMDP either differentiates to the granulocytic lineage or matures into the MDP, which is capable of differentiation to monocytes/macrophages and to the CDP, which is a committed progenitor of the dendritic cells. Abbreviations: HSC: hematopoietic stem cell; MPP: multipotent progenitor; LMPP: lymphoid-primed multipotent progenitor; CMP: common myeloid progenitor; MLP: multi-lymphoid progenitor; ETP: early T-cell progenitor; MEP: megakaryocyte-erythrocyte progenitor; GMDP: granulocyte/monocyte/dendritic progenitor: MDP: monocyte/dendritic progenitor: CDP: common dendritic progenitor.

**Figure 2 ijms-23-10529-f002:**
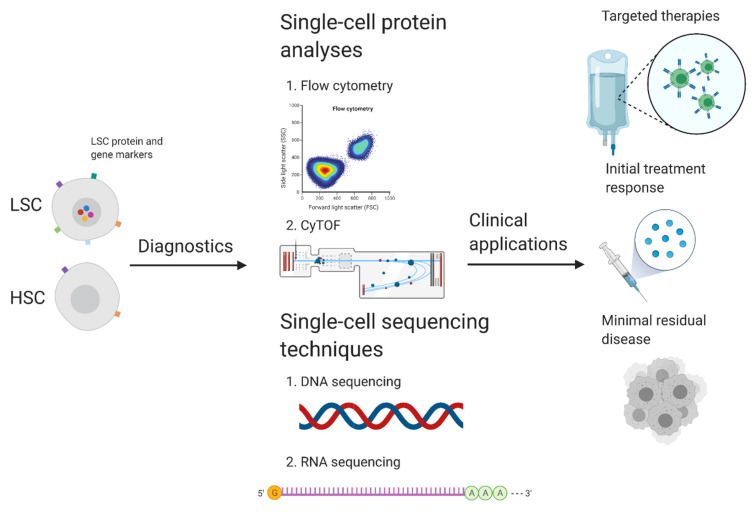
Potential implications of distinguishing normal and leukemic blast cells based on protein and/or gene profiles. Distinguishment of normal and leukemic blast cells based on their protein and/or gene expression profile can be performed by single-cell protein analysis techniques, such as flow cytometry and CyTOF, and/or single-cell sequencing of their DNA or RNA profile. Combining these techniques to detect (therapy specific) protein or gene markers can lead to the development of novel targeted therapy approaches, monitoring of initial treatment response, and detection of minimal residual disease to prevent relapse.

**Table 1 ijms-23-10529-t001:** Overview of the different CD markers that are positive or negative on the different subtypes of hematopoietic stem cells and myeloid progenitor cells.

Cell Type	CD34 CD38 Compartment	Positive Markers				Negative Markers			
**CD34−HSC**	CD34−CD38−	CD90+	CD133+	GPI−80+		CD45RA−	CD110−	CD135−	
**CD34+ HSC**	CD34+CD38−	CD44+	CD90+	CD133+	CD135+	CD3−	CD10−	CD14−	CD19−
		GPI−80+				CD45RA−	CD56−	CD66b−	CD335−
**MPP**	CD34+CD38−	CD44+	CD133+	CD135+		CD3−	CD10−	CD11c−	CD14−
						CD19−	CD45RA−	CD49f−	CD56−
						CD66b−	CD90−	CD335−	GPI−80−
**CMP**	CD34+CD38+	CD123int	CD135+			CD3−	CD10−	CD11c−	CD14−
						CD19−	CD45RA−	CD49f−	CD56−
						CD66b−	CD90−	GPI−80−	
**MEP**	CD34+CD38+	CD123int	CD133low			CD3−	CD10−	CD11c−	CD14−
						CD19−	CD45RA−	CD56−	CD66b−
						CD90−	CD135−	GPI−80−	
**GMDP**	CD34+CD38+	CD33+	CD45RA+	CD117+	CD135+	CD3−	CD10−	CD11c−	CD14−
		HLA−DR+				CD19−	CD56−	CD66b−	CD90−
						CD115−	CD116−	CD123−	GPI−80−
**MDP**	CD34+CD38+	CD33+	CD45RA+	CD115+	CD117+	CD3−	CD10−	CD11c−	CD14−
		CD123hi	CD135+	HLA−DR+		CD19−	CD56−	CD66b−	CD90−
						CD116−	CD123−	GPI−80−	
**CDP**	CD34+CD38+	CD11c+	CD33+	CD45RA+	CD116+	CD3−	CD10−	CD11c−	CD14−
		CD117+	CD123hi	CD135+	HLA−DR+	CD19−	CD56−	CD66b−	CD90−
						CD115−	GPI−80−		

Abbreviations: HSC: hematopoietic stem cell; MPP: multipotent progenitor; CMP: common myeloid progenitor; MEP: megakaryocyte-erythrocyte progenitor; GMDP: granulocyte/monocyte/dendritic progenitor: MDP: monocyte/dendritic progenitor: CDP: common dendritic progenitor.

## Data Availability

Not applicable.

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
