# Peer review of "Flow Cytometric Identification of Hematopoietic and Leukemic Blast Cells for Tailored Clinical Follow-Up of Acute Myeloid Leukemia"

_ijms, 2022, doi:10.3390/ijms231810529_

Round 1

Reviewer 1 Report (Previous Reviewer 2)

This review describes the expression of common markers evaluable by flow cytometry used in diagnosis of AML, and compares their expression in benign and leukemic myeloid blasts, eventually useful to define MRD.

Minor revisions:

Authors should check typos and some error throughout all the review.

CD90 is mentioned as marker that remains positive in CD34+CD38- fraction, but is the first time that it is mentioned in the text, while in table 1 is listed. Same thing for CD44 in MPP.

Between the GMDP negative markers, authors reported also CD1. They should specify if they referred to a specific subunit (CD1a, CD1b, …) or to a pan-reactive anti-CD1. They should also specify that CD135 is similarly expressed to CD34+HSC.

In the sentence 226 authors should remove “intermediately” because they explain the expression level immediately after.

Table 1 as to be revised because some markers are not properly listed and some population are not in the right CD34/CD38 cells compartment.

Between aberrant marker expression should be considered also CD19.

The reference reported at line 327 (44) does not mention % of CLL-1 expression in AML cases.

In conclusion section, author should also consider that marker expression profile may change also due to therapy in the same patient.

Author Response

Reviewer 2 Report (New Reviewer)

This review summarizes recent relevant papers and provides comprehensive information in terms of flow cytometry markers that are useful to distinguish normal and AML blasts. The manuscript is well written and organized. I have no criticism on it. 

Author Response

Reviewer 3 Report (New Reviewer)

- The paragraph 4.”distinction between healthy hematopoietic- and leukemic blast cells”  is confusing and put together numerous concepts in addition to the criteria of distinction. My suggestion is to add sub-paragraphs in which to discuss: a)single-cell techniques, in light of the characterization between normal and leukemic blasts, which in the current form are barely mentioned only in Figure 2 and in lines 420-421; b) therapeutic implications that the improvement of discrimination between healthy and leukemic blast cells could entail.

- In chapter 4, in the part describing conjugated mAbs, it might be interesting to add a brief description of the treatment with CD33 conjugated antibody (GO) for CD33+ patients, that is the first targeted therapy based on the surface antigen expression.

Minor comments:

Figure 1: Abbreviations are lacking in the caption.

Line 412: ALOT: add reference.

Round 2

Reviewer 3 Report (New Reviewer)

The authors addressed all my comments, and the manuscript is now well organized and more clear. 

This manuscript is a resubmission of an earlier submission. The following is a list of the peer review reports and author responses from that submission.

Round 1

Reviewer 1 Report

Overall, I just think that the structure of the paper is not of a systematic review. It followed the PRISMA guidelines for article search but it did not followed the structure to be reported as one. It lacks some necessary protocol details. There are also several conceptual problems. Including, you cannot say that AML is characterized by high proliferation and accumulation of LSCs, like HSC, these are rare cells, and AML is made of mostly blasts; leukemic transformation can occur in any point of progenitor differentiation, not specifically in HSC; and LSCs are not the majority of leukemic cells in the disease. They should have detailed as inclusion criteria the phenotype that they would consider as LSCs, so it would be clearer. Perhaps they should say that the purpose was to find differences between normal and leukemic progenitors, not specifically LSCs. The findings and conclusion are useful to explore new, unusual markers to follow up AML though. Anything that can be related to prognosis or used to detect MRD and predict relapse, clinically is worth taking a look. 

Over concerns: 
- Why only 2 data bases? Typically at least three databases are used for a systematic review. Did you check Cochrane database for similar systematic reviews?
- Are all the keywords used Mesh terms?
- There is a lack of protocol information on methodology. This is important in a systematic review. How did they access quality of individual studies? Did they use strobe? Did they do cohen’s kappa statistic to as a measurement of Inter-Rater Reliability? Did they registered the protocol on PROSPERO database? Which method was used to access risk of bias (regression test?)? All of this are criteria to a good systematic review.
- Data extraction and management should be in methods. Saying specifically which information they would get from the reviewed papers. A systematic review consist in a SYSTEMATIC way to get data from everything that you can find in literature on one subject to answer a specific question, so you have to tell on methodology which specific data were collected from the papers.
- The first thing that I missed in results in the classical overview table with the characteristics of the studies included in the final analysis.
- The description of aberrant markers on CD34+ CD38- population. But this is not useful to identify or differentiate LSCs from HSCs. Because they are aberrant and possibly found on blats and not necessarily LSCs.
- Figure 3: Velten, Nat Comm, 2021 - some good data already published.

Reviewer 2 Report

In this review Weeda and co-author explore the available literature to dissect immunophenotype of the human hematopoietic progenitors and stem cells, comparing it with leukemic cells phenotype, in order to highlight differential markers helpful to distinguish healthy from malignant cells in AML and therefore useful in MRD.

Major revision:

It seems to me that the authors use LSC, in the introduction and consequently in several passages of the manuscript, as a synonymous for either the true Leukemic Stem Cells, that initiate this malignancy and from which blasts develop, and the leukemic bulk (Leukemic Cells, or blasts, or…) that comprise the cells that effectively accumulate provoking HSC expulsion from BM: “It is characterized by the accumulation of abnormal and non-functional (differentiated) stem cells, so-called leukemic stem cells (LSCs). In AML, hematopoietic stem cells  (HSCs) have undergone leukemic transformation, induced by their mutational background. The uncontrolled proliferation and accumulation of LSCs consequences in expulsion of healthy hematopoietic cells in the BM.” This concept is well addressed in reference 44 of bibliography.

Due to this, the paper as to be profoundly revised from author to properly term the two different populations.

Minor revisions:

Authors had to revise the bibliography, because some citation is not correct.

Introductions:

Lines 36-37: The word “differentiated” in brakets seem to suggest that differentiation means non-functionality.

At lines 76-78, the authors stated that Euroflow panel “do not include markers to identify leukemic blast cells and therefore do not identify whether stem cell subsets and progenitors are healthy or leukemic”. I think that could be more appropriated to assert that “this kind of panel can identify only the general leukemic blast population, without distinguish the different subsets of progenitors”. By the way, the definition of healthy or leukemic with flow cytometry is usually possible only because of an expansion of the blast compartment (at the onset) or if aberrant markers are co-expressed by blasts. Even identifying the maturative step represented could be not helpful in assessing healthness or not. It is of paramount importance, as authors properly underline, to identify markers that are not express in HSC at any level of their hierarchy, while understand from which level of this branching blasts develops, is ancillary for the patient management.

Methods:

Lines 109-116: publications not in English was quite obvious, being English publication only an inclusion criterion. I suggest merging this section with the successive (lines 114-115, therefore excluding  lines 112-113). I also suggest removing sentence at line 116.

Figure 1. Asterisks have to be removed from PRISMA flow diagram because they do not report to anything; in “identification” step, could be interesting to split databases record in Pubmed records and Google Scholar records. In “included” section are reported n=70 studies and n=70 reports, but  previously n=38 reports assessed for eligibility via “identification of studies via other methods” was reported. Was a typo or I misunderstood? At line 122 was better to avoid three times repetition of “searching”?

Results:

Lines 154-174: this paragraph has to be more fluent, avoiding several repetition and short sentences.

Line 181: I could not find any reference about CD38 expression in article 16 of bibliography. Maybe authors referred to 32.

Line 183: References 30 and 31 reports data on CD34+ population, here authors were described CD34- fraction

Lines 191-192: CD135 and CD133 are inverted.

Lines 202-203 what about CD133 and GPI80 expression? They were deregulated? And CD44?

Lines 210-218: I suggest inserting a sentence to highlight that from this maturation step all the subsequent myeloid progenitor cells belong to the CD34+CD38 compartment, in order to remove the repetition at the beginning of every following paragraph (i.e. lines 221, 229, 237, 247).

Line 239: CD115 and CD117 are inverted.

Lines 242-243: some markers are repeated and CD123 is missing (cited in table 1).

Line 248: authors listed CD116 as a marker already expressed by MDP, but either in text and in table is in not expressed marker list.

Line 249: CMP instead of CDP.

Line232: “observed on CD34+ HSCs”, because the CD34- counterpart does not express it.

Table 1. I suggest to list CD markers in ascending order and left aligned to increase order and readability. Reference 32 in bibliography reports CD90+ expression also in CD34+ HSC, and reference 30 detects CD133 expression in MPP and CD133low expression in MEP. Could this point be deepened?

Why was not reported also LMPP and MLP marker expression? Could it be more appropriate to modify the caption  with “haematopioetic stem and myeloid progenitor cells”? Authors have to decide if to use haematopoietic or hematopoietic.

Paragraph 3.2 and Table 2: It is not clear to me the usefulness to report also the markers commonly expressed in healthy HSCs (i.e. CD33, CD44, CD45RA, CD123, CD135), since it will not be helpful in MRD evaluation. Frequent sentence interruption that make discourse less fluid (particularly from line 275 to 300) should be avoided.

Line 329: CD56 is not an activation marker of T cell, but identify a distinct T cell subpopulation (https://doi.org/10.3389/fimmu.2017.00892 as an example).

Lines 342-344: CD99 is also characteristic of hematogones in BM, so is not correct to assert “CD99 is not expressed on HSCs and progenitor cells, which allows separation of LSCs and healthy hematopoietic stem- and progenitor cells”.

Lines 391-395: Probably an “and” is missing in this sentence “is mostly expressed in the cytoplasm of normal  hematopoietic blast cells and granulocytes”. Moreover, I can not find any information about “overexpression” in papers cited.

Lines 408-412: besides listed markers at line 408, authors missed CD44, that erroneously mention at line 412. Moreover, I do not agreed with the insertion of CD99 at line 412, as mentioned above.

Lines 418-420: It is not clear to me the reason why the authors decide to suggest these panels of marker combinations to distinguish LSC from HSCs and what is the rationale. In fact, they propose markers that do not cover the 100% of expression frequency on LSC, along with marker usually expressed also by HSCs. This point  has to be elucidated or suggestions should be removed.

Line 420: Typo (à instead of a).

Line 445: success.

Lines 450-452: Same than above.

Authors cite some ongoing studies and therapy strategy that take advantage of aberrant markers expression of cancer cells. I think that this interesting section should be enriched with more literature. Someone had already published some therapy trial that uses markers emerged and mentioned in authors review?

Lines 449-450: “Therefore, a combinatorial targeting approach has been proposed in which multiple antigens are targeted with CAR T-cells” This sentence needs some reference.

Reviewer 3 Report

The manuscript ”Flow cytometric distinguishment of hematopoietic and leukemic stem cells for tailored clinical follow-up of acute myeloid leukemia: a systematic review” by Weeda et al. is a review article on the utility of flow cytometry and characterizing normal and leukemic hematopoietic stem cells (HSC) in the diagnosis and follow-up of patients with acute myeloid leukemia (AML).  The topic has practical utility and has been addressed by various other groups. 

The manuscript may benefit from addressing the following questions and suggestions, prior to further consideration:

  1. General comments: The manuscript is confusing and difficult to follow, primarily due to incorrect and/or inconsistently used nomenclature.  The authors appear to use interchangeably the terms “blasts / myeloid blasts (or myeloblasts)” and “stem cells”, whether normal or leukemic.  This is not accurate, and examples are detailed below. 

  2. Title: We suggest replacing “distinguishment” with “identification”, as a more commonly used term for English language readers. 

  3. Introduction: This section contains a number of partially correct or inaccurate statements related to AML.  For example, AML is defined in the 2017 WHO classification as an increase in myeloid blasts (or myeloblasts), typically exceeding 20%, rather than leukemic stem cells. 

    The authors also state that “in the last 30 years, therapies for AML have not been tailored to the individual patient level for the majority of the AML subtypes”.  In fact, there were a number of new drugs or drug combinations approved in the last 5-10 years for the personalized treatment of patients with AML, including FLT3 inhibitors such as midostaurin and gilterinib; the combination of hypomethylating agents and venetoclax (HMA+VEN); liposomal daunorubicin/cytarabine formulation for the treatment of t-AML and AML-MRC; and IDH1/IDH2 inhibitors such as ivosidenib and enasidenib. 

    The authors then indicate that “…allogeneic stem cell transplantation is recommended after treatment in young patients or patients with a reduced risk of treatment related death.”  This is a partially correct statement, as the indication of alloSCT is based on the risk stratification of AML patients, based primarily on cytogenetic and/or molecular abnormalities. 

    The sentence “The appearance of new healthy hematopoietic stem- and progenitor cells during and after chemotherapy treatment is often confused with the presence of residual leukemic cells, since these cells morphologically display (almost) identical characteristics.” is not accurate.  Stem cells cannot be identified by microscopic assessment.  The earliest myeloid progenitor cells that can be identified during routine peripheral blood or bone marrow morphologic evaluation is the myeloid blast.

    The sentence “Next to this, the heterogeneous mutation background of AML prevents monitoring by the current state-of-the-art molecular diagnostics.” is not accurate.  The 2018 European LeukemiaNet guidelines on MRD assessment in AML, also updated in 2021 (PMID: 34724563) clearly state the evidence-based utility of molecular residual disease evaluation in patients with AML and recurrent abnormalities, including acute promyelocytic leukemia, core binding factor AML, and AML with mutated NPM1.

  4. Results and Discussion: The incorrect and interchangeable use of “leukemic stem cells” and “myeloid blasts” lead to erroneous statements like “A subsequent study confirmed by Chen et. al. confirmed the low frequency of CD2 expression on LSCs [48].”  The reference cited (and other ones) clearly refer to leukemic myeloid blast immunophenotyping and aberrant marker expression, not leukemic stem cells.
  5. Language and Grammar:  The manuscript has fairly extensive grammatical errors and would benefit from an English language copyediting service.
